# Acidic enol electrooxidation-coupled hydrogen production with ampere-level current density

Zheng-Jie Chen[1,5], Jiuyi Dong[1,5], Jiajing Wu[2,5], Qiting Shao[1], Na Luo[1], Minwei Xu[1], Yuanmiao Sun [1], Yongbing Tang [1,3], Jing Peng[1,3] ✉ & Hui-Ming Cheng [1,3,4] ✉

Hydrogen production coupled with biomass upgrading is vital for future sustainable energy developments. However, most biomass electrooxidation reactions suffer from high working voltage and low current density, which substantially hinder large-scale industrial applications. Herein, we report an acidic hydrogen production system that combined anodic ascorbic acid electrooxidation with cathodic hydrogen evolution. Unlike C-H and O-H bonds cleavage with slow kinetics in conventional organic oxidation, the highly active enol structure in ascorbic acid allows for an ultralow overpotential of only 12 mV@10 mA/cm² using Fe single-atom catalysts, and reaches 1 A/cm² at only 0.75 V (versus reversible hydrogen electrode) with approximately 100% Faraday efficiency for hydrogen production. Furthermore, the fabricated two-electrode membrane-free electrolyser delivers an industrial current density of 2 A/cm²@1.1 V at 60 °C (2.63 kWh/Nm³ H₂), which requires half of the electricity consumption in conventional water electrolysis (~5 kWh/Nm³ H₂). This work provides a new avenue for achieving industrial-scale hydrogen production from biomass.

Hydrogen has recently garnered great attention as a promising clean and renewable energy source for sustainable development and carbon neutrality[1–4]. Water electrolysis, powered by sustainable electricity, is an economically and environmentally friendly approach for producing hydrogen[5–7]. Unfortunately, the efficiency is greatly limited by oxygen evolution reaction (OER) process with sluggish kinetics, thereby leading to a high voltage (e.g., >1.6 V) to achieve meaningful H₂ production[8,9]. It is reported that about 90% of the energy consumption for electrolytic water is stemmed from OER contribution[10], and the membrane is necessary to obstruct gas crossover forming explosive H₂/O₂ mixtures. However, the reactive oxygen species during electrolysis will exacerbate the degradation of the membrane and shorten membrane life, thus further increasing the cost of water

electrolyzer[11,12]. Therefore, developing hydrogen production systems coupled with anodic reactions that are thermodynamically more favorable, safer, cost-efficiency than OER is highly desirable.

Recently, a particularly appealing system has been proposed to replace OER with electrooxidation of biomass and its derivatives (such as alcohols, glycerin, and 5-hydroxymethylfural, glucose, etc.), which show sustainable nature and will not disrupt the current carbon balance of our ecosystem[13–17]. Their electrooxidation has a theoretical potential lower than OER, and produces upgraded oxidative species[18–22]. Consequently, water electrolysis coupled with these anodic reactions not only conserves electricity consumption, but also accelerates hydrogen production and high-value by-product generation rather than low-valued O₂ in OER process. Nevertheless, current

¹Faculty of Materials Science and Energy Engineering/Institute of Technology for Carbon Neutrality, Shenzhen Institute of Advanced Technology, Chinese Academy of Sciences, Shenzhen 518055, China. ²Institute of Information Technology, Shenzhen Institute of Information Technology, Shenzhen 518172, China. ³Shenzhen Key Laboratory of Energy Materials for Carbon Neutrality, Shenzhen Institute of Advanced Technology, Chinese Academy of Sciences, Shenzhen 518055, China. ⁴Shenyang National Laboratory for Materials Science, Institute of Metal Research, Chinese Academy of Sciences, Shenyang 110016, China. ⁵These authors contributed equally: Zheng-Jie Chen, Jiuyi Dong, Jiajing Wu. ✉e-mail: jing.peng@siat.ac.cn; hm.cheng@siat.ac.cn

electrooxidation reactions such as alcohols and aldehydes typically require the application of potentials greater than 1.23 V and operate at current densities lower than 200 mA/cm² [23–25]. In this case, biomass electrooxidation is accompanied by a competitive OER process that needs costly membranes and thus gives rise to high system cost, low efficiency and high energy consumption of hydrogen production. This is likely ascribed to high overpotentials in biomass electrooxidation processes, which involve high-energy carbon-hydrogen (C-H) and oxyhydrogen bond (O-H) breaking (Fig. 1). The slow kinetics and membrane resistance result in large working potentials, necessitating high energy input to overcome.

Enol is a structure that encompasses a hydroxyl group covalently bonded to a carbon atom of a double bond. In comparison to C-H and O-H bonds in alcohols or aldehydes, enol structure is more reactive and its hydrogen atom is more susceptible to dissociation. Conceivably, biomass with enol structure is expected to realize faster kinetic electrooxidation and efficient hydrogen production with low overpotential and applied potential, so as to circumvent the need for a membrane and lower the material/operation cost. In this work, we report the utilization of ascorbic acid (AA), which has an enediol group containing hydroxyl groups on both sides of the double bond [26], as an anode additive to replace OER for hydrogen production. AA is a natural product with enol structure that can be readily synthesized by biological fermentation [27], and its oxidation product, dehydroascorbic acid (DHA), is used in medical, cosmetic and pharmaceutical industries [28,29]. Owing to the reductive enediol group, AA has a theoretical oxidation potential of 0.48 V (versus reversible hydrogen electrode (RHE)) [30] that is far lower than the OER, making it an appealing alternative to replace the OER. Furthermore, oxidation of AA to DHA involves only two-electron transfer and is kinetically more advantageous than other organic molecules [31].

By combining the merits of AA, an acid, an efficient and membrane-free system for hydrogen production together with AA electrooxidation was developed. Resulting from the highly active enol structure, the overpotential in AA oxidation reaction (AAOR) lowers to only 12 mV@10 mA/cm² using Fe single-atom catalysts, and the current density reaches to 1 A/cm² at 0.75 V (vs. RHE). Product analysis indicates that only DHA produces in anodic electrode and the cathode product H₂ has a Faraday efficiency of 100%. In addition, the two-electrode electrolyzer exhibits only 1.1 V cell voltage to achieve an industrial current density of 2 A/cm² at 60 °C, and the electricity consumption is approximately half of the conventional water electrolysis (~5 kWh /Nm³). Theoretical studies have demonstrated that the strong adsorption of Fe³⁺ with HA⁻ anions leads to a strong p-π conjugation effect of C-O in the enol structure, which promotes the dissociation of the hydroxyl group in the enol structure. This work provides guidance for the design of low-cost, efficient and safe hydrogen production systems from biomass electrooxidation.

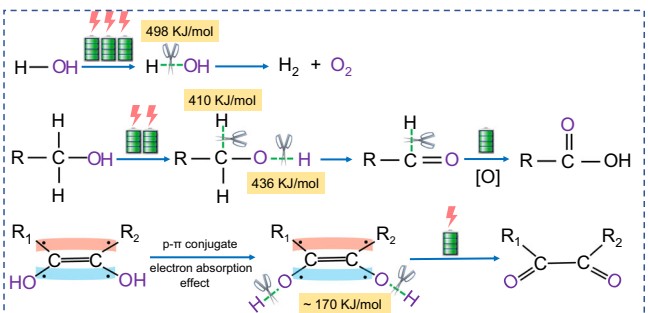

**Fig. 1 | Schematic diagram of the dissociation energy of C-H bond and O-H bond in different molecular structures.** The scissors and the green dashed line refer to the possible broken bonds of the oxidation process and the dissociation energy of the corresponding bonds.

## Results and discussion

### Synthesis and structural characterization of catalysts

Single-atom catalysts are a prevailing electrocatalytic materials used in various catalytic reactions because of high atomic utilization and fantastic catalytic activity [32,33]. Given that the autoxidation of AA by oxygen in the presence of transition metals, especially ferric (Fe(III)) and cupric (Cu(II)) ions [34,35], we designed and synthesized Fe single-atom catalysts anchored on Ketjen black ($x$%Fe@KJ, where $x$ is the mass loading) with different mass loadings ranging from 1 to 10 wt.% by a calcination method (Supplementary Fig. S1). From aberration-corrected scanning transmission electron microscopy (AC-STEM) images in Fig. 2a, the 2.5% Fe@KJ catalyst shows a hollow vesicle structure. Many isolated Fe (Co, Ni, Cu) atoms were identified by the magnified image (Fig. 2b and Supplementary Fig. S2), which indicates the single-atom character in 2.5%Fe(Co, Ni, Cu)@KJ catalyst. From the high-angle annular dark field (HAADF) image and energy-dispersive X-ray spectroscopy (EDS) mapping, we found that Fe, O, and C elements in the KJ present a homogeneous distribution (Fig. 2c). In addition, we also performed high-resolution TEM (HRTEM) characterization of catalysts with different Fe loadings, as shown in Supplementary Figs. S3 and S4. For catalysts with different Fe contents, both Fe and O are uniformly distributed in the KJ support. With the increase of Fe loadings, Fe gradually transforms from single-atom dispersion to nanoparticles. Moreover, the higher the Fe loadings, the larger amounts of nanoparticles. X-ray diffraction analysis for 2.5%Fe@KJ showed the only peak of carbon (Supplementary Fig. S5), further demonstrating that Fe mixed with KJ presented atomic-level dispersion.

X-ray absorption fine structure spectroscopy (XAFS) was performed to verify the chemical states and the atomic dispersion of Fe species with precise coordination structure. As seen in Fig. 2d, the K-edge X-ray absorption near-edge structure (XANES) spectrum of 2.5%Fe@KJ exhibits a profile that closely resembles that of Fe₂O₃ reference, thereby verifying the trivalent oxidation state of Fe atoms in 2.5%Fe@KJ. Fourier transforms (FTs) of XANES further reveal a dominant peak at ~1.5 Å, which is attributed to the Fe-O bond (Fig. 2e), while the absence of a peak at approximately 2.2 Å signifies the non-existence of Fe-Fe bonds in 2.5%Fe@KJ. This finding is in agreement with STEM results and confirms the single-atom dispersion of Fe with oxygen-atom coordination. To further distinguish the coordination information of Fe, an extended XAFS (EXAFS) wavelet transform analysis was performed (Fig. 2g–i). We found that 2.5%Fe@KJ shows a maximum intensity at ~5.0 Å⁻¹ and Fe atom is the single nucleation center. Subsequently, the EXAFS curve-fitting analysis was carried out to extract the quantitative coordination structure for the Fe moiety in 2.5%Fe@KJ (Fig. 2f and Supplementary Fig. S6, and Supplementary Table 1). These analyses revealed that the coordination number of Fe in 2.5%Fe@KJ was estimated to be 5 in the first shell, with the Fe-O distance of 2.1 Å. In the fore-edge region, a weak peak at ~7113 eV was observed in the 2.5%Fe@KJ spectrum, which is mainly due to the charge transfer from the 1s orbit of O ligand to 3d orbit of Fe metal. These results suggest that the ligand geometry around Fe is a square structure, as this fore-edge peak is usually considered as a fingerprint of a square planar with a porphyrin-like configuration [36].

### Electrochemical performance test of catalysts

Traditional Fe-based catalysts play a vital role in numerous electrocatalytic reactions [37,38]. In our case, despite of an acidic AA solution at pH = 2.3, the 2.5%Fe@KJ catalyst exhibits an outstanding AA oxidation performance that allows for an ultralow overpotential of 12 mV at a current density of 10 mA/cm², greatly outperforming the OER performance with the acidic benchmark OER catalyst IrO₂. Conversely, when the electrolyte is H₂SO₄ with the same pH, the current density of 2.5% Fe@KJ catalyst is almost negligible below 2 V, demonstrating that the anodic current in the AA solution is a result of AAOR rather than OER (Fig. 3a). Many biomass electrooxidation reactions have been reported

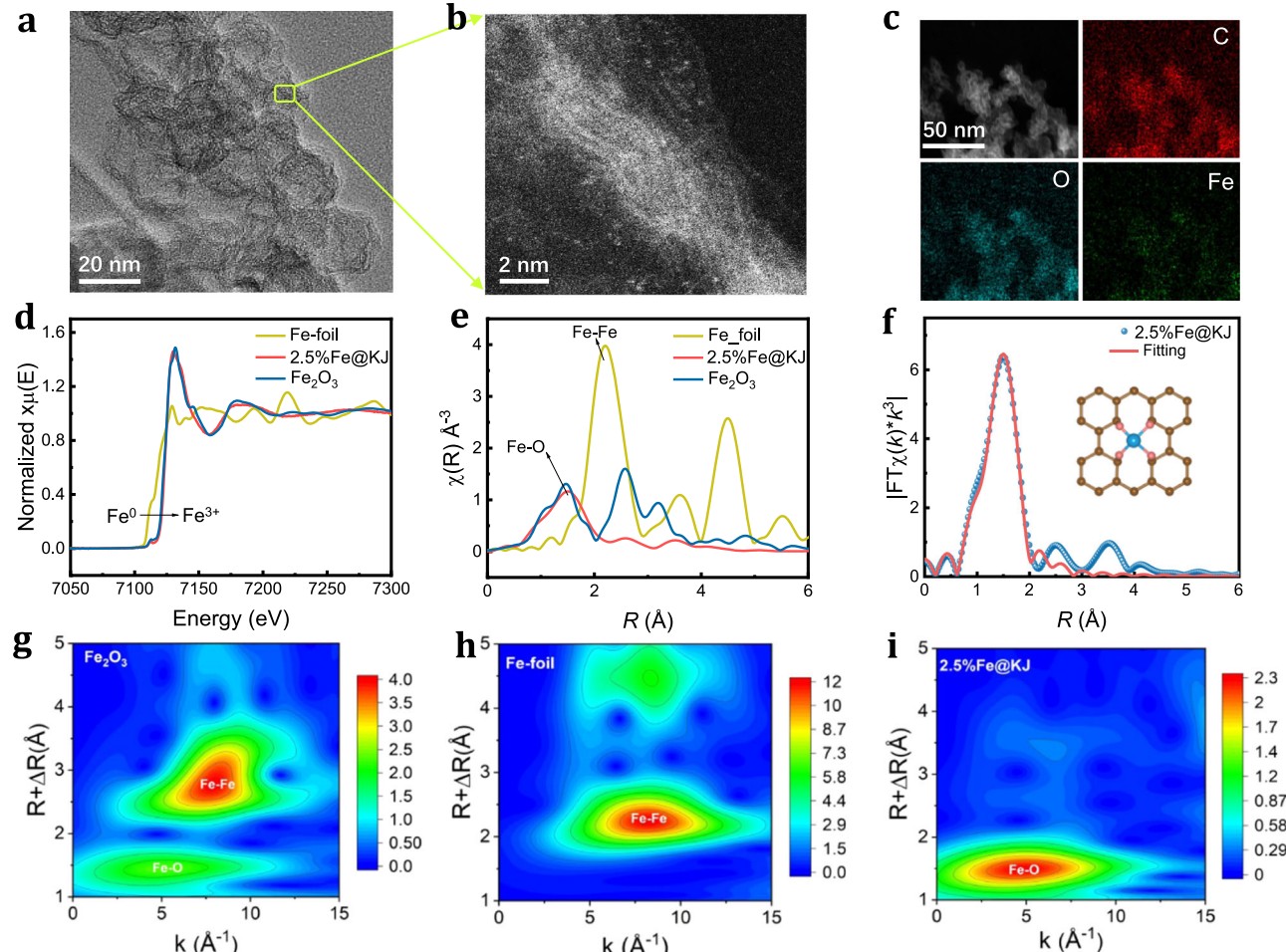

**Fig. 2 | Structure characterizations by AC-HAADF-STEM and XAFS. a, b** HAADF-STEM images at different magnifications. **c** Corresponding EDS maps. **d** Fe K-edge XANES spectra of 2.5%Fe@KJ and reference samples. **e** Fe K-edge EXAFS signal of 2.5%Fe@KJ and reference samples. **f** Fe K-edge EXAFS signal (blue sphere) and curvefit (pink line) for 2.5%Fe@KJ. **g–i** Wavelet transforms for the k3-weighted EXAFS signals of 2.5%Fe@KJ and reference samples. The color bar represents the intensity of the elemental bonding peaks.

for hydrogen production; nonetheless, the majority of them have an overpotential of more than 600 mV (Fig. 3b), which results in large working potential in anodic half-reaction, even exceeding 1.23 V. The ultralow overpotential in AAOR thus requires only 0.492 V to achieve a current density of 10 mA cm$^{-2}$. This is primarily attributed to the p-π conjugation effect that enhances the bias of the oxygen electron cloud towards the delocalized bond, making it easier to dissociate the H from the hydroxyl group[39], thereby requiring only a low potential to drive AA oxidation.

Subsequently, the AAOR performance by various metal-based (Fe, Co, Ni, Cu) catalysts with identical mass loadings was evaluated. We found that 2.5%Fe@KJ catalyst exhibited the best catalytic performance, where it achieved an ultra-high current density of 1 A/cm$^2$ at a potential of only 0.75 V (vs. RHE) at room temperature (RT) (Fig. 3c). Conversely, the AAOR performance catalyzed by pure KJ was significantly inferior to that of metal-based catalysts, implying that transition metals served as the active centers for catalytic AA oxidation. To determine the optimal loading, the AAOR performance of catalysts with varying Fe contents@ KJ was further tested, and we found that the ideal Fe wt.% loading (@KJ) was approximately 2.5 wt.% (Supplementary Fig. S7). In order to detect the real activity, we also calculated the double-layer capacitance ($C_{dl}$) and electrochemical surface area (ECSA) for different metal and Fe loadings (Supplementary Figs. S8 and S9). It was found that the $C_{dl}$ of different metal single-atom catalysts were around 15 mF cm$^{-2}$, and the corresponding ECSA

were about 255. The close $C_{dl}$ indicates that the capacitance of the single-atom catalyst is contributed by KJ and carbon paper. Furthermore, the LSV curve of normalized ECSA showed that the 2.5% Fe@KJ single-atom catalyst exhibited better $j_{ECSA}$ than the other catalysts. In addition, the LSV curve of normalized ECSA of different Fe contents also revealed that 2.5% Fe@KJ presented better catalytic activity than the other loadings.

Electrochemical impedance spectroscopy (EIS) and Bode plot were recorded with different potentials to investigate the kinetic process of AAOR (Fig. 3d and Supplementary Fig. S10). Notably, in the low-frequency region, the Nyquist plot exhibits a dramatic change as the potential increased, gradually splitting from a straight line to two semicircular curves when the potential exceeded 0.6 V vs. RHE, indicating the occurrence of AAOR. As the operating potential increases, the semicircular curve of AAOR becomes small, which suggests a decrease in impedance and faster reaction kinetics of AAOR. Moreover, EIS of various metals (Supplementary Fig. S11) show that the charge transfer impedance of Fe single-atom catalyst is the smallest at the same potential compared to other metals, indicating faster kinetics of 2.5%Fe@KJ catalyst, which is consistent with the LSV results. The long-term stability of catalysts under acidic conditions is a significant challenge, particularly for non-precious metal electrocatalysts. The durability of 2.5%Fe@KJ catalyst for AA oxidation was evaluated by chronopotentiometry assays at 100 mA/cm$^2$. The results indicated that 2.5%Fe@KJ catalyst activity did not exhibit significant degradation for

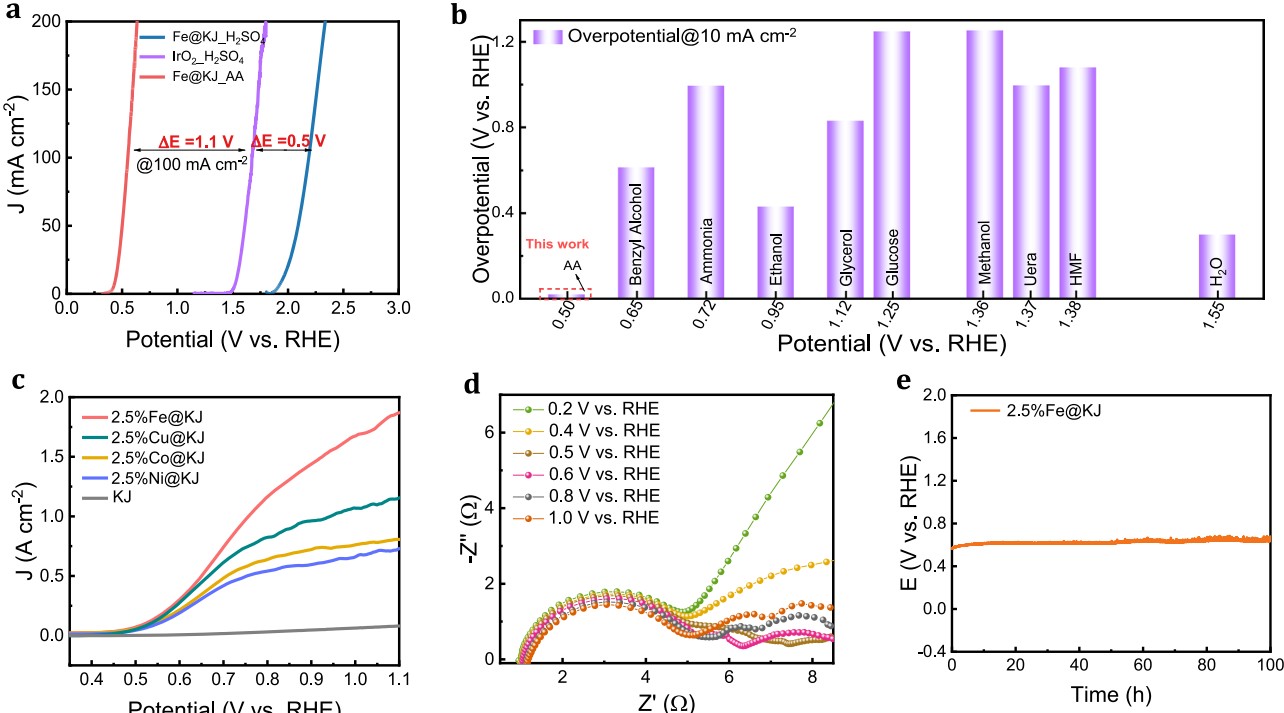

**Fig. 3 | Electrochemical performance. a** Potential difference ΔE of Fe@KJ in AA and H₂SO₄ solution at pH = 2.3. **b** Comparison of overpotential of different anode additives[10, 20, 46–48]. **c** LSV curves with scan rate of 5 mV/s of AAOR with different metal@KJ in 1 M AA solution. **d** Nyquist plots of different potential for 2.5%Fe@KJ. **e** Electrochemical stability of 2.5%Fe@KJ catalyst for 100 h in 1 M AA solution.

more than 100 h (Fig. 3e), surpassing the long-term stability of most currently reported hydrogen production system coupled with biomass electrooxidation. Besides, the morphology and structure change of catalysts after 100 h durability tests was characterized by AC-STEM. From Supplementary Fig. S12, we found that 2.5%Fe@KJ still exhibits atomic-level dispersion, which demonstrated that the catalyst shows long-term stability. In addition, Fe ions in AA solution were measured at different reaction times by ICP-OES, revealing that the corrosion rate of Fe ions was approximately 0.0042 ug/(L h) (Supplementary Fig. S13), indicating that 2.5%Fe@KJ manifests excellent stability.

**Product characterization of AA by electrocatalytic oxidation**
Infrared spectroscopy (IR) and nuclear magnetic resonance (NMR) were performed to investigate the evolution of the anode products and their concentrations over time. As we can see from the IR spectrogram, the newly generated carbonyl group (-C=O) peak (assigned to the peak of 1797 cm⁻¹) emerges and becomes evident with increasing reaction time. In contrast, the peak of C=C group gradually decreases, suggesting that AA was oxidized to the intermediate products HA⁻ (ionization of a proton of AA) and DHA (Fig. 4a, b). For the qualitative and quantitative analysis of AA reaction products, we performed 4-h chronoamperometry (i-t) measurements, collecting liquids at 1-h intervals. The collected liquids were characterized by ¹H NMR. Before the reaction, we found three distinct wrapped peaks in the NMR spectrum of AA mainly at -4.92, -4.05, and -3.7 ppm, respectively. As the reaction time increased, we found three new wrapped peaks (peaks 1,2,3) appeared, and comparing them with the standard peaks of DHA (Supplementary Fig. S14), we concluded that these three wrapped peaks were DHA peaks. Meanwhile, we observed that the peak intensity of AA was weakening as the reaction time increased. Quantitative analysis revealed that the yield of DHA could reach -87% after 4 h at a voltage of 0.75 V vs. RHE (Fig. 4c). The above results indicate that the product of AA is mainly DHA, which is consistent with the results of the IR spectrum.

**Mechanism of AA reaction by electrocatalytic oxidation**
Structural characterization using aberration-corrected HAADF-STEM and EXAFS fit analysis revealed that the Fe atoms in KJ are atomically dispersed with square planes resembling a porphyrin-like structure[36]. Subsequently, a density functional theory (DFT) model was constructed with the predominant (M=Fe, Ni, Co, Cu)-O-C bonding mode (Supplementary Fig. S15). The adsorption energy of HA⁻ on various single-atom metals (M-O-C) was calculated (Supplementary Fig. S16), and HA⁻ adsorption on FeO₆ was found to be more stable than on other metals. This observation indicates that FeO₆ single atoms have a greater affinity for HA⁻. The hybridization orbitals of the metal elements with O in HA⁻ showed that Fe has stronger hybridization with O, suggesting that Fe³⁺ is more favorable for capturing HA⁻ (Supplementary Fig. S17). In addition, electron localization function diagram further confirmed the order of the adsorption capacity of each metal as: Fe > Cu > Co > Ni, which agrees with experimentally measured performance results (Supplementary Fig. S18).

To further rationalize the superior activity of Fe-O single-atom catalysts, Gibbs free energy diagrams were calculated for the oxidation of AA to DHA for four different metal single atoms at U = 0 V and 1 V (Fig. 4d and Supplementary Fig. S19). Previous research has demonstrated that -OH away from the carbonyl group in the enol structure is easily hydrolyzed (kpa1 = 4.79) to form stable HA-, whereas -OH near the carbonyl group is weakly ionized (kpa2 = 11.75), thus impeding deprotonation[40]. We propose a possible mechanism with the above analysis. We first take the stable HA⁻ as the initial state as shown in Fig. 4e. Next, an electrophilic adsorption process occurs between the metal cation, followed by HA⁻ anion deprotonation and electron loss. Finally, a process of metal and DHA desorption takes place. Calculations revealed a negative Gibbs free energy for step I for all metals, signifying that this process can be spontaneous. The Gibbs free energy diagram exhibits that step II is the rate-determining step for all metals oxide catalysts. The C, O, and H electron transfer in the enol-like structures of HA⁻ and metal-adsorbed HA⁻ were analyzed using the

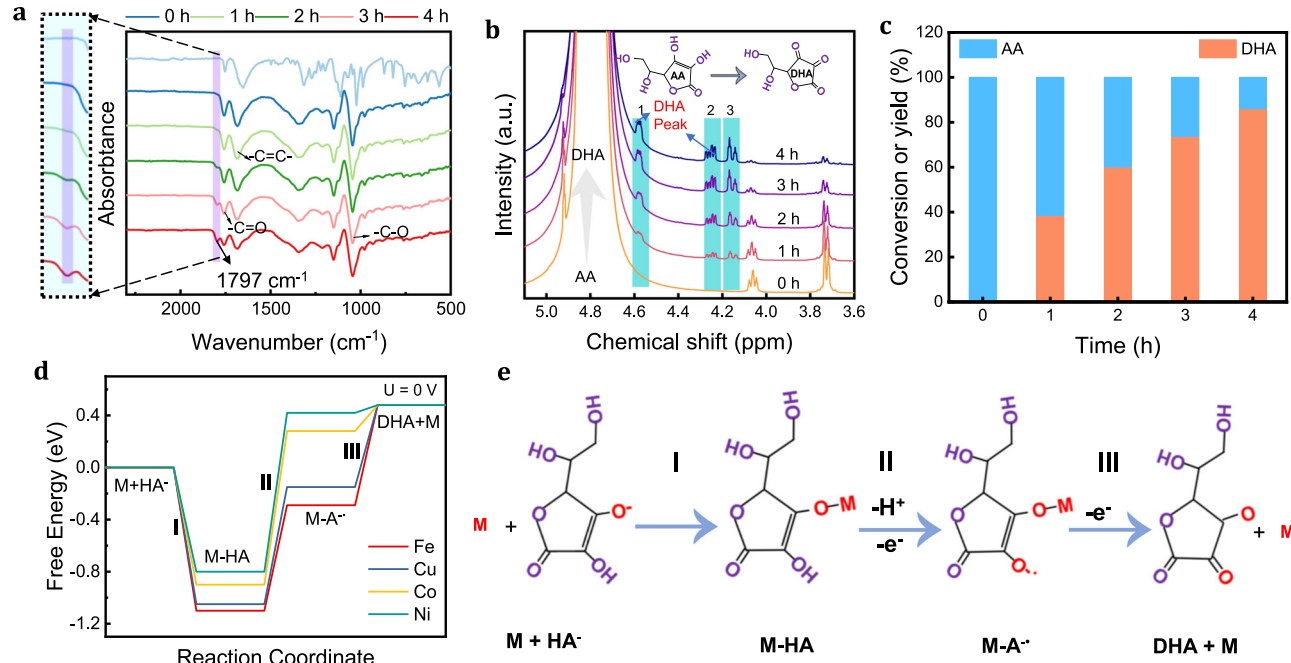

**Fig. 4 | Characterization of AA oxidation products. a** IR spectrum and **b** NMR spectrum for AA with different oxidation times in 1 M AA solution. **c** Yield rates of DHA at different reaction times. **d** Gibbs free energy diagram for different metals at a potential of U = 0 V. **e** Mechanism of AA reaction. M(Fe, Cu, Co, Ni) and HA⁻ refer to metal and the dissociation of a hydrogen from AA, respectively. M-HA presents HA⁻ adsorbed on metal. M-A·˙ means M-HA lost a proton and an electron.

Bader charge calculation method (Supplementary Figs. S20 and S21). We discovered that the charge transfer of H in the enol structure of HA⁻ adsorbed with metals was higher than that in the initial HA⁻, indicating a stronger polarization of -OH with metal catalysis. Specifically, the amount of charge transfer of the adsorbed HA⁻ on the $FeO_8$ single-atom catalyst is larger than the other three metal oxide single-atom catalysts, demonstrating that the $FeO_8$ catalyst more readily promotes the dissociation of the hydroxyl group on the other side of the enol structure. These results indicate that the stronger adsorption of metal cations with HA⁻ anions results in a stronger p-π conjugation effect of C-O in the enol structure, leading to a greater polarization of the hydroxyl group on the other side. Therefore, $MO_8$ single atoms effectively reduce the energy barrier of HA⁻ deprotonation.

## Energy consumption of two-electrode system

To explore the feasibility of utilizing AAOR-coupled systems for hydrogen production, a membrane-free two-electrode electrolyzer was constructed in which 2.5%Fe@KJ was utilized as the anode catalyst and Pt mesh was selected as the cathode catalyst for hydrogen evolution (Fig. 5a, b). Polarization curves were evaluated at 25 and 60 °C to determine the system performance. The results indicated that only 1.5 V was required to achieve a current density of 2 A/cm² at 25 °C with the two-electrode system. At 60 °C, a current density of 2 A/cm² was achieved at a mere 1.1 V and a current density up to 4 A/cm² could be attained at a cell voltage of 1.5 V (Fig. 5c). Furthermore, faraday efficiency of the hydrogen product was calculated to be 100%, indicating no O₂ or CO₂ products (Fig. 5d). Moreover, the ¹H NMR characterization of the anode product in the flow cell after 12 h showed that the yield of DHA could reach 98.3%. From Fig. 4b, c and Supplementary Fig. S22, the peak of DHA becomes more obvious as the reaction time increases and no other product peaks appear, indicating that the remaining species are mainly unreacted AA. Compared with traditional hydrogen production techniques such as water electrolysis and biomass electrooxidation, the AAOR-coupled electrolyzer displayed lower cell voltage and electricity expense, especially at large current density. Specifically, electrical energy consumption was around 2.63 kWh/m³ H₂ at 2 A/cm², which is nearly half of the energy consumption of water

electrolysis (Fig. 5e). These findings suggested that AAOR-coupled water electrolysis held great potential for practical applications in the future.

In summary, we have developed an effective and acid-based membrane-free system for the electrooxidation of AA in parallel with hydrogen production. Due to the highly reactive enol structure present in AA, the overpotential for AAOR is only 12 mV @ 10 mA/cm² when using 2.5%Fe@KJ single-atom catalyst, which allows a current density of 1 A/cm² to be achieved at 0.75 V (vs. RHE) in the anodic half-reaction. Notably, this system exhibits a Faradaic efficiency of approximately 100% and demonstrates long-term stability of over 100 h for hydrogen production. Utilizing a two-electrode electrolyzer, a cell voltage of just 1.1 V was sufficient to achieve an ampere-level current density of 2 A/cm² at 60 °C, and only requires a low electricity consumption of 2.63 kWh/Nm³ H₂. This research not only provides a promising technique for the cost-effective and safe production of H₂ as well as the upgrading of biomass, but also establishes a foundation for exploring valuable enols (such as phenol and catechol that have a low price and higher industrial value) electrooxidation-coupled electrolysis systems or self-co-electrolysis[41,42].

## Methods

### Materials

Iron sulfate hydrate (Fe₂(SO₄)₃, AR, Fe 21–23%) and anhydrous sodium sulfate (Na₂SO₄, ACS, ≥99.0%) purchased from Shanghai Aladdin Biochemical Technology Company. AA (C₆H₈O₆, 99.99%), Nafion 117 perfluorinated resin solution, cobalt chloride hexahydrate (CoCl₂•6H₂O, 99.99%), copper chloride dihydrate (CuCl₂•2H₂O, AR), nickel chloride, hexahydrate (NiCl₂•6H₂O, 99.9%) was purchased from Shanghai Maclean Biochemical Technology Company. Cogent Black (ECP-600JD) Purchased from Cloride Sodium borohydride (NaBH₄, ≥97%) Purchased from Shanghai Lingfeng Chemical Reagent Company Limited.

### Preparation of iron-loaded Ketjen black catalyst (Fe@KJ)

Ketjen black (KJ, 90 mg) and iron sulfate hydrate (Fe₂(SO₄)₃• xH₂O, 71 mg), were dissolved in deionized water (30 ml) and mixed ultrasonically for 30 min to make a mixture and set aside. Weigh sodium

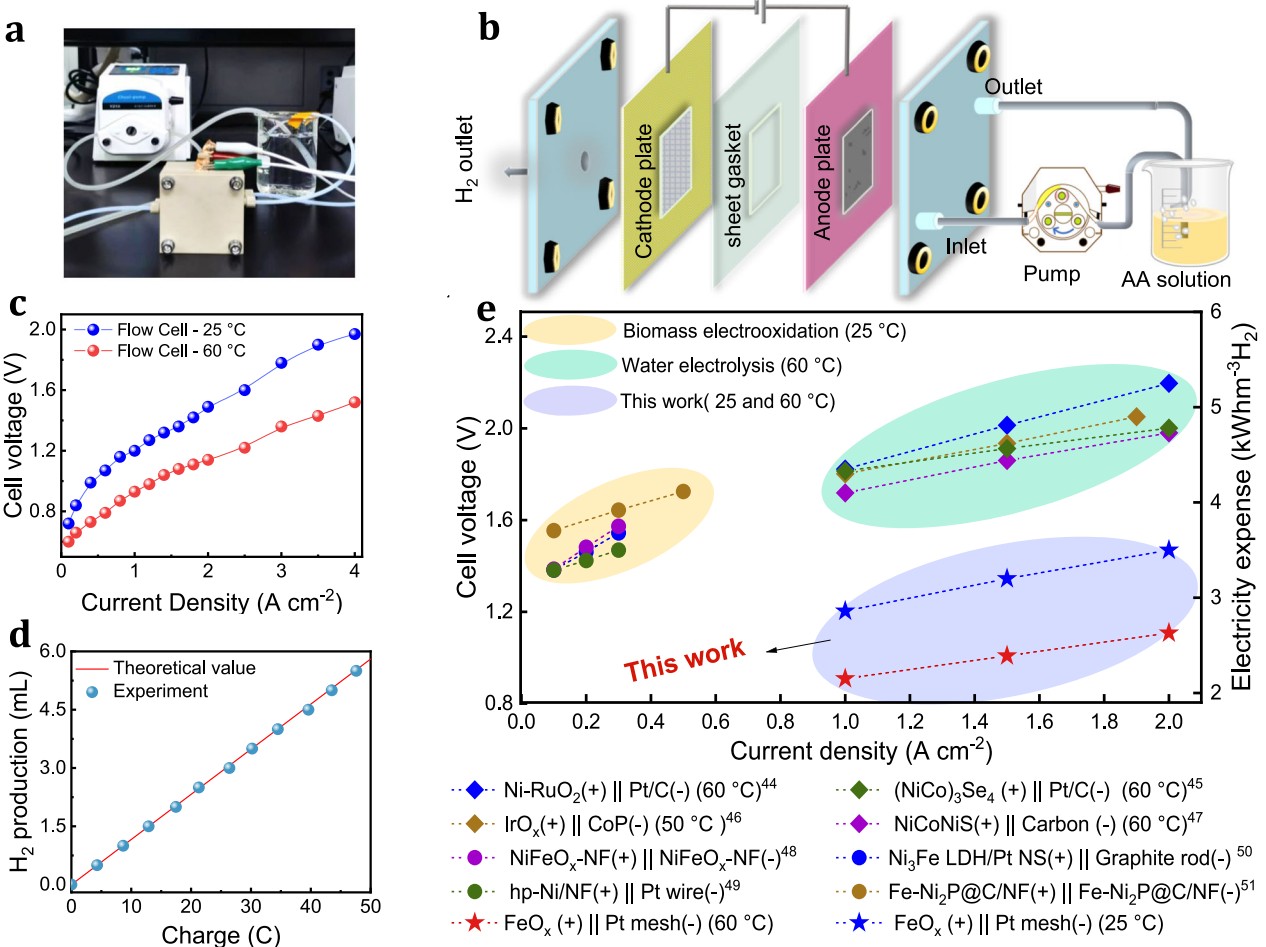

**Fig. 5 | Performance of two-electrode system. a** Physical picture of flow cell. **b** Schematic diagram of the internal structure of flow cell. **c** Polarization curves for flow cell at 25 and 60 °C in 1 M AA solution. **d** Faraday rate of hydrogen production with 1 M AA solution. **e** A comparison of electricity expense for hydrogen production with water electrooxidation[49–52] and biomass electrooxidation[53–56] systems. Orange, green and blue regions represent electrooxidation for biomass, water and AA.

borohydride (NaBH₄, 100 mg) and dissolve in 10 ml of deionized water. Stir the mixture while slowly adding NaBH₄ solution dropwise and react for 2 h. Transfer into a centrifuge and centrifuge at 10000r for 10 min, pour off the supernatant, then wash twice with water and once with ethanol, respectively, put the remaining mixture into a vacuum oven and set the temperature at 80 °C to dry for 12 h. When the tube furnace temperature reaches 350 °C, the dried powder sample is quickly pushed into the tube furnace and annealed for 2 h under a mixed hydrogen-argon atmosphere (5% H₂/Ar). Allow the samples to cool naturally to room temperature before collection.

In the synthesis of other catalysts, only the ratio of metal precursors (mass fraction of Fe elements) or replacement of metal precursors was needed to synthesize by the same method. M-KJ (M=Co, Ni, Mn, Cu) was synthesized in the same way as Fe@KJ. Cobalt chloride hexahydrate (CoCl₂•H₂O), copper chloride dihydrate (CuCl₂•2H₂O), nickel chloride, and hexahydrate (NiCl₂•6H₂O) were used as metal precursors.

### Preparation of working electrodes
The catalyst (5 mg) prepared in the above procedure was weighed and dissolved in a mix solution of isopropyl alcohol (0.75 ml) and deionized water (0.25 ml). Then add 50 ul of Nafion solution, sonicate the mixed solution for 1 h, and finally take 20 µl of the mixture with microsampler, apply it on hydrophilic carbon paper (the area of

catalyst coating on carbon cloth is 0.15 cm²) and dry it with an infrared lamp to prepare the working electrode.

### Material characterization
The morphology and microstructure of the catalysts were analyzed by field emission SEM (Thermo Scientific Phenom Pharos PW-100-160), TEM (FEI Tecnai G2 spirit) and aberration-corrected HAADF-STEM (Titan Themis Z) equipped with EDS. Powder sample X-ray diffractograms were recorded on a Rigaku diffractometer (MiniFlex 600) with Cu Kα radiation (20 kV, λ = 0.154056 nm). A PerkinElmer Frontier FTIR spectrometer was used for in situ FTIR testing, and synchrotron X-ray absorption measurements were performed on a SPring-8 X-ray absorption beamline at Harima Science Garden City, Hyogo). Data reduction, data analysis, EXAFS fitting and wavelet transform were performed with Athena, Artemis and HAMA Fortran version software packages, respectively.

### Preparation of electrolyte
AA (7 g) and sodium sulfate (Na₂SO₄, 5.6 g) were dissolved in deionized water (40 ml), sonicated until completely dissolved, and then configured into a solution of AA and Na₂SO₄ with a concentration of 1 M.

### Electrochemical measurement
All electrochemical tests for AA oxidation were performed in 1 M Na₂SO₄ solution on a CHI 660E workstation. The Ag/AgCl and Pt mesh

were used as the reference electrode and counter electrode, respectively. Measurements of three-electrode system were carried out in a membrane-free electrolyzer. The two-electrode system was tested at room temperature and 60 °C in an electrolyzer, and LSV curves were obtained in the potential range of −0.2 to 0.8 V vs. AgCl at a scan rate of 5 mV s$^{-1}$. The resistance compensation was 80%. All RHE potentials were from Eq: $E(vs.\ RHE) = E(vs.\ AgCl) + 0.0591*pH$ ($pH$ of 1 mol AA is 2.3) $+ 0.1989$. $H_2$ is obtained by the gas collection method of drainage.

$C_{dl}$ and ECSA electrochemical tests. First, we selected a non-Faraday voltage window (−0.05 to 0.05 V vs. Ag/AgCl) and performed CV cycling tests in this voltage range with sweep rates of (20, 40, 60, 80, 100, 120 mV/s). Next, points were taken at the potentials of 0 V vs. Ag/AgCl and the sweep speed was plotted as a function of the corresponding current density. Finally, the function is fitted linearly to find the slope of the function, which is also known as $C_{dl}$, and then ECSA can be calculated by $C_{dl}$.

The following equation was applied to measure ECSA of the electrodes at different cell voltages.

$$C_T = I/(dE/dT) \quad (1)$$

$$ECSA = C_T/C*A \quad (2)$$

where $C_T$ is the total capacitance (F), $I$ is the current (A), $dE/dt$ is the voltage scan rate (V s$^{-1}$), $C*$ is the specific capacitance (F cm$^{-2}$), $A$ refers to area of electrode and $ECSA$ is the electrochemical surface area per geometric area (cm$^2$ cm$^{-2}$). The main limitation of this method is the general assumption that the specific capacitance of oxides is 60 μF cm$^{-2}$, regardless of the oxide composition and crystalline structure, and without considering the composition of the electrolyte in which the measurement is carried out.

## Calculation of energy efficiency and conversion rate

The Faradaic efficiency of all the products were calculated based on their corresponding electron transfer per molecule oxidation using the following equations.

$$FE(\%) = mole_{experimentally\ product}/mol_{theoretically\ product} \times 100 \quad (3)$$

$$mol_{theoretically\ product} = Q/(n \times F) \quad (4)$$

where $mol_{theoretically\ product}$ is the mole number of the product. $Q$ is the transfer charge, $n$ is the number of electrons transferred for products and $F$ is Faraday's constant (96,485 C mol$^{-1}$).

$$Yield\ Conversion(\%) = (mol_{Product}/(mol_{Product} + mol_{Reactants})) \times 100 \quad (5)$$

Where $mol_{Product}$ is the molarity of the products. $mol_{Reactants}$ is the molarity of the Reactants.

The electricity consumption per m$^3$ of $H_2$ produced was calculated as follows:

$$W = (n \times F \times U \times 1000)/(3600 \times V_m) \quad (6)$$

where $n$ is the molarity of electrons transferred for product, $U$ is the input voltage and $V_m$ is the molar volume of the gas at standard atmospheric pressure (22.4 mol l$^{-1}$). $F$ is Faraday's constant. It is noted that $n = 2$ for organic electrooxidation-coupled a cathode-only hydrogen production system.

## Product characterization

The solution $^1$H NMR was carried out on a 400 MHz NMR spectrometer (Bruker AV400) to quantitatively identify the AA oxidation products at different times. The product solutions used for the NMR test were obtained at a potential of 0.75 V vs. RHE. The solutions of AA oxidation products of different times measured by IR were extracted at a current density of 10 mA cm$^{-2}$.

## Theoretical calculation details

The DFT calculations were carried out by the Vienna ab initio simulation package (VASP)[43] and the vaspkit tool was employed to process the calculation results[44]. The exchange-correlation was described with the Perdew-Burke-Ernzerhof functional at the generalized gradient approximation level[45]. The cut-off energy was set as 450 eV, and gamma-centered $k$-point meshes were constructed for all structures in this work. The convergence e threshold of energy tolerance of $1.0 \times 10^{-5}$ eV per atom and force tolerance of $1.0 \times 10^{-2}$ eV/Å during the geometry optimization. We calculated the Gibbs free energy of the AA oxidation process based on the standard hydrogen electrode model.

The adsorption energy ($E_{ads}$) of AA was calculated with the following equation:

$$E_{ads} = E_{Fe*DHA} - E_{Fe} - E_{DHA} \quad (7)$$

Where $E_{Fe*DHA}$, $E_{DHA}$, $E_{Fe}$ are the total energy of different models.

The steps of the oxidation of AA to DHA:

$$HA^- + M \rightarrow M - HA^- \quad (8)$$

$$M - HA^- \rightarrow M - HA^{-\bullet} + e^- + H^+ \quad (9)$$

$$M - HA^{-\bullet} \rightarrow M + DHA + e^- \quad (10)$$

Since AA is an enol structure, the H in an OH of enol in water is more easily hydrolyzed, step I (Eq. (8)) of the reaction starts with HA$^-$ as the adsorbent. It is noted that step I (Eq. (8)) is the process of electrophilic adsorption, step II (Eq. (9)) is the process of deprotonation and loss of electrons, and step III (Eq. (10)) is the process of loss of electrons and desorption.

The change in Gibbs free energy (ΔG) can be expressed as follows:

$$\Delta G = \Delta E_{ads} + \Delta ZPE - T\Delta S - eU - 0.0591^* pH \quad (11)$$

where $\Delta ZPE$, $T\Delta S$ is the difference of zero-point energy and mixed entropy. When calculating the step diagram of the Gibbs free energy change, all metal-catalyzed AA take the first step as a benchmark to calculate the electron-gaining process. The effects of pH and electrode potential are considered at each step of the calculation, which can be treated with terms "0.059*pH" and "eU", respectively.

## Reporting summary

Further information on research design is available in the Nature Portfolio Reporting Summary linked to this article.

# Data availability

The main data supporting the results of this study can be found in the article and its supplemental materials. Source Data are provided with this paper or obtained from figshare repository at https://doi.org/10.6084/m9.figshare.23575293.

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

## Acknowledgements

The work was financially supported by the National Natural Science Foundation of China (Nos. 52188101; H.-M.C.; 22275205; J.P. and 22205148; J.W.), Shenzhen Basic Research Project (No. JCYJ20200109144616617; H.-M.C.), the Science and Technology Foundation of Shenzhen (No. JCYJ20220530154404010; J.P.), Guang Dong Basic and Applied Basic Research Foundation (No. 2023B1515020102; J.P. and 2022A1515110408; Z.-J.C.), China Postdoctoral Fund (No. 2022M713270; Z.-J.C.) and Cross Institute Joint Research Youth Team Project of SIAT (E25427; J.P.). The computing work associated with this paper was supported by the public computing service platform provided by SIAT.

## Author contributions

H.-M.C., and J.P. initiated and designed this project. Z.-J.C., J.D. and J.W. synthesized and characterized the samples, tested the performance and performed the theoretical calculations. Z.-J.C., J.P. and H.-M.C. co-wrote the manuscript. Q.S., N.L., M.X., Y.S., and Y.T. participated in the analysis of data. All authors contributed to the discussion of the results and comments on the manuscript.

## Competing interests

The authors declare no competing interests.
