## [Peer Review File · Nature Communications]

REVIEWER COMMENTS

Reviewer #1 (Remarks to the Author):

In this work, the authors report an acidic hydrogen production system that combine anodic ascorbic acid electrooxidation with cathodic hydrogen evolution. Owing to the highly active enol structure, the ascorbic acid oxidation reaction exhibits lower overpotential using Fe single-atom catalysts. Product analysis indicates that there is only dehydroascorbic acid in anode and H₂ in cathode, and the Faraday efficiency of H₂ is 100%. In addition, the energy consumption of this hydrogen production system is approximately half of conventional water electrolysis, which could also obtain more valuable product in anode. Besides that, such an acidic hydrogen production system might be able to overcome (partially) some persistent issues in water electrolysis, which is meaningful. However, some problems in the current work should be addressed before this work can be considered for publication.

- 1) In the introduction, the authors should mention the technical challenges of employing such a system for hydrogen production and biomass upgrading. By mentioning the challenging and also providing possible solutions, the significance of this work can be further improved.
- 2) The authors have listed the problem of oxygen evolution reaction (OER) and the energy consumption of water electrolysis stemmed from OER. Besides, the use of membrane and the different kinetics of anode and cathode reactions may also bring some problems. It is suggested to add some discussion on it and cite the recent related reports for this point, such as *Angew. Chem. Int. Ed.* 2023, e20230356, and *Adv. Energy Mater.* 2023, 13, 2203455.
- 3) From the physical characterization it is clear that the 2.5%Fe@KJ catalyst is a Fe single-atom catalyst. However, other catalysts with different mass loadings of Fe are not given corresponding characterizations. If the authors want to call all these catalysts of single-atom catalysts, corresponding characterizations should be provided.
- 4) What does the x represent (line 98-99 in page 4)? Is the amount of Fe precursor or the actual amount of Fe in the catalysts? How did the authors confirm the actual mass loading of Fe?
- 5) The data in Fig. 3d and Fig. S5 are non conclusive. From the description in the text, when the potential exceeded 0.6 V vs. RHE, the Nyquist plot in the low-frequency region gradually splits from a straight line to two semicircular curves. However, this is not obvious in the Figures. I suggest that the authors either re-collect this plot or show difference plots or something to make the differences more obvious.
- 6) Is this catalyst stable under 60 °C system? Could the authors prove this?
- 7) In Fig. 4a, the peak of carbonyl group (1800 cm⁻¹) is not consistent in the text (1797 cm⁻¹). Please confirm this.

Reviewer #2 (Remarks to the Author):

This manuscript describes the study of the electrooxidation of acidic enol with hydrogen production. The study highlights the potential for reducing the process power use to within an ampere of current density, which is a significant accomplishment. This potential breakthrough is of broad scientific and social interest because of its implications for clean hydrogen production and renewable energy use.

Electrooxidation commonly requires voltages of 0.5 to 3.0 V with currents of 2 to >100 mA/cm². This study claims to achieve ascorbic acid electrooxidation with 12 mV and 10 mA/cm² with an almost 100% Faraday efficiency for hydrogen production. It also claims to consume half the electricity than water electrolysis at an industrial scale. These values are much lower than those found in the public literature.

The results provide visual and analytical evidence for the performance of the Fe-based catalyst. The authors provide reasonable descriptions for the interpretation of the X-ray diffraction and XAFS analyses. The corrosion rate of Fe ions result is sufficient, if limited, evidence of catalyst stability. Additional stability analysis could be considered beyond the scope of this study.

The overpotentials shown in Figure 3b are a strong highlight of the potential significance of this study. These results suggest that AA electrooxidation could be a better resource for hydrogen production than water electrolysis. This observation must be confirmed with a lifecycle assessment that includes the production of AA.

Figure 5e shows another significant result. The study shows that the electricity expense for hydrogen production is much lower than biomass electrooxidation and water electrolysis. It also shows that electricity consumption decreases with increasing operating temperature.

The conclusions could expand upon the potential implications of this work. For example, are there other organic compounds that could achieve similar results, given the new understanding from this study? Are there other considerations related to AA that should be investigated, such as cost or resource availability?

Reviewer #3 (Remarks to the Author):

This is an interesting and important work that reported the electrochemical oxidation of ascorbic acid (AA) to dehydroascorbic acid (DHA) and pairing it with hydrogen evolution reaction (HER) at the cathode for cogeneration of valuable chemical and H₂. The research includes electrochemical in-situ FTIR, and DFT, as well as flow cell tests. It can be published in Nature Communication after addressing the following comments.

1. As we know, Fe³⁺ and AA can react to form DHA without a catalyst. Whether the 2.5%Fe@KJ reported by author can react with AA to produce DHA without electricity? Can the reaction between AA and Fe can selfpower the HER process like *Advanced Materials*, 2022, 34, 2200058; *Angewandte Chemie International Edition* 2023, e202218603.

2. What is the stability of 2.5%Fe@KJ catalyst? The author should report the morphology and structure change of catalysts after 100 h durability tests which may also important to understand the reaction mechanism.

3. The yield of DHA is 87%. What are the other anode products?

4. The quantitative analysis of anode products in the flow cell is suggested to be provided.

5. More electrochemical tests, such as ECSA, and so on, are suggested to be provided to solid the conclusion.

6. The size and format of the picture in the article are not uniform. The unit of current density is wrong (Figure 3a) and so on.

7. Recent references on the electrooxidation of small organic molecules coupled with HER should be cited in this paper, especially under acidic conditions. Such as, *Angewandte Chemie International Edition* 2021, 60, 21464–21472.

Point-by-Point Responses to Reviewers' Comments

We are truly grateful to the reviewer's valuable comments and suggestions on our work, which has greatly improved the quality and clarity of this manuscript. All comments and suggestions have been taken into account in the revised manuscript, as described below.

Response to Reviewer #1:

In this work, the authors report an acidic hydrogen production system that combine anodic ascorbic acid electrooxidation with cathodic hydrogen evolution. Owing to the highly active enol structure, the ascorbic acid oxidation reaction exhibits lower overpotential using Fe single-atom catalysts. Product analysis indicates that there is only dehydroascorbic acid in anode and H₂ in cathode, and the Faraday efficiency of H₂ is 100%. In addition, the energy consumption of this hydrogen production system is approximately half of conventional water electrolysis, which could also obtain more valuable product in anode. Besides that, such an acidic hydrogen production system might be able to overcome (partially) some persistent issues in water electrolysis, which is meaningful. However, some problems in the current work should be addressed before this work can be considered for publication.

Author's response:

We thank the reviewer for acknowledging the importance of our work. We have carried out supplementary experiments and have made modifications according to the reviewer's constructive and valuable suggestions.

1) In the introduction, the authors should mention the technical challenges of employing such a system for hydrogen production and biomass upgrading. By mentioning the challenging and also providing possible solutions, the significance of this work can be further improved.

Author's response:

Thanks to reviewer for the valuable suggestions. We have revised the introduction section, where corresponding revisions are marked in blue and displayed as follows:

“In this case, the biomass electrooxidation is accompanied with a competitive OER process that needs costly membranes, and thus gives rise to high system cost, low efficiency and high energy consumption of hydrogen production.”

“The slow kinetics and membrane resistance result in large working potentials, necessitating high energy input to overcome.”

“Conceivably, biomass with enol structure is expected to realize faster kinetic electrooxidation and efficient hydrogen production with low overpotential and applied potential, so as to circumvent the need for a membrane and lower the material/operation cost.”

2) The authors have listed the problem of oxygen evolution reaction (OER) and the energy consumption of water electrolysis stemmed from OER. Besides, the use of membrane and the different kinetics of anode and cathode reactions may also bring some problems. It is suggested to add some discussion on it and cite the recent related reports for this point, such as *Angew. Chem. Int. Ed.* 2023, e202303563, and *Adv. Energy Mater.* 2023, 13, 2203455.

Author’s response:

We truly appreciate the reviewer’s helpful suggestions. We have added some discussion on the use of membrane and cite the recent related reports as follows:

“ It is reported that about 90% of the energy consumption for electrolytic water is stemmed from OER contribution¹⁰, and the membrane is necessary to obstruct gas crossover forming explosive H₂/O₂ mixtures. However, the reactive oxygen species during electrolysis will exacerbate the degradation of membrane and shorten membrane life, thus further increase the cost of water electrolyzer^{11,12}.”

The cited references as follows:

11 Wu, K., Li, H., Liang, S., Ma, Y. & Yang, J. Phenazine-based Compound Realizing Separate Hydrogen and Oxygen Production in Electrolytic Water Splitting. *Angew. Chem. Int. Ed.*, e202303563 (2023).

12 Ma, Y., Wu, K., Long, T. & Yang, J. Solid-State Redox Mediators for Decoupled H₂ Production: Principle and Challenges. *Adv. Energy Mater.* 13, 2203455 (2023).

3) From the physical characterization it is clear that the 2.5%Fe@KJ catalyst is a Fe single-atom catalyst. However, other catalysts with different mass loadings of Fe are not given corresponding characterizations. If the authors want to call all these catalysts of single-atom catalysts, corresponding characterizations should be provided.

Author’s response:

Thank you for your precious comments and advices. We have provided corresponding characterizations of TEM images for other catalysts with different mass loadings of Fe as well as other metal single-atom catalysts, as shown in Figs.R1-R3. For catalysts with different Fe contents, both Fe and O are uniformly distributed in the carbon support (Figs. R1 and R2). With the increase of Fe content, Fe gradually transforms from single-atom dispersion (Fig. 2b) to nanoparticles. Moreover, the higher the Fe content, the larger amounts of nanoparticles. In addition, other catalysts with different metal were characterized by AC-HAADF-STEM. All of these catalysts with 2.5% content exhibit atomic-level dispersion (Fig. R3). Therefore, they are called single-atom catalysts.

Fig. R1 HRTEM images and mapping images for 10%Fe@KJ catalysts.

Fig. R2 HRTEM images and mapping images for 5%Fe@KJ catalysts.

Fig. R3 AC-HAADF-STEM images for different metal single-atom catalysts.

4) What does the x represent (line 98-99 in page 4)? Is the amount of Fe precursor or the actual amount of Fe in the catalysts? How did the authors confirm the actual mass loading of Fe?

Author's response:

We thank the reviewer for the insightful question. $x\%Fe@KJ$, the x is the amount of Fe precursor content, which ranges from 1 wt.% to 10 wt.%. By weighing the mass of $Fe_2(SO_4)_3$, the actual Fe content is obtained by stoichiometry ratio conversion, so that x here refers to a theoretical loading content.

5) The data in Fig. 3d and Fig. S5 are non conclusive. From the description in the text, when the potential exceeded 0.6 V vs. RHE, the Nyquist plot in the low-frequency region gradually splits from a straight line to two semicircular curves. However, this is not obvious in the Figures. I suggest that the authors either re-collect this plot or show difference plots or something to make the differences more obvious.

Author's response:

We thank the reviewer for the insightful questions and valuable suggestions. In order to make the differences of the Nyquist plots more obvious, we have re-collect the curve at potential of 0.5 V vs. RHE (Fig. R4). When the potential is greater than 0.5 V vs. RHE, the low-frequency region is mainly limited by mass transfer, and the Nyquist plot in the low-frequency region gradually splits from a straight line into approximately two semicircular curves, indicating that the occurrence of AAOR at 0.5 V vs. RHE. The re-collected plots have been updated in the revised manuscript.

Fig. R4 Nyquist plots of different potential for 2.5%Fe@KJ.

6) Is this catalyst stable under 60 °C system? Could the authors prove this?

Author's response:

Thanks for the reviewer's comments. By testing the stability of 2.5%Fe@KJ catalyst at 60 °C with two electrode system, it was found that its stability could last for more than 100 hours at a current density of 100 mA cm⁻² (Fig. R5), indicating that the catalyst was highly stable in the flow cell.

Fig. R5 Stability test of 2.5%Fe@KJ in a flow cell with two-electrode system.

7) In Fig. 4a, the peak of carbonyl group (1800 cm⁻¹) is not consistent in the text (1797 cm⁻¹).

Please confirm this.

Author's response:

We sincerely appreciate the reviewer for notifying us of typological errors. The corresponded text has been corrected to be 1797 cm⁻¹.

Response to Reviewer #2:

This manuscript describes the study of the electrooxidation of ascorbic acid with hydrogen production. The study highlights the potential for reducing the process power use to within an ampere of current density, which is a significant accomplishment. This potential breakthrough is of broad scientific and social interest because of its implications for clean hydrogen production and renewable energy use.

Electrooxidation commonly requires voltages of 0.5 to 3.0 V with currents of 2 to >100 mA/cm². This study claims to achieve ascorbic acid electrooxidation with 12 mV and 10 mA/cm² with an almost 100% Faraday efficiency for hydrogen production. It also claims to consume half the electricity than water electrolysis at an industrial scale. These values are much lower than those found in the public literature.

The results provide visual and analytical evidence for the performance of the Fe-based catalyst. The authors provide reasonable descriptions for the interpretation of the X-ray diffraction and XAFS analyses. The corrosion rate of Fe ions result is sufficient, if limited, evidence of catalyst stability. Additional stability analysis could be considered beyond the scope of this study.

The overpotentials shown in Figure 3b are a strong highlight of the potential significance of this study. These results suggest that AA electrooxidation could be a better resource for hydrogen production than water electrolysis. This observation must be confirmed with a lifecycle assessment that includes the production of AA.

Figure 5e shows another significant result. The study shows that the electricity expense for hydrogen production is much lower than biomass electrooxidation and water electrolysis. It also shows that electricity consumption decreases with increasing operating temperature.

The conclusions could expand upon the potential implications of this work. For example, are there other organic compounds that could achieve similar results, given the new understanding from this study? Are there other considerations related to AA that should be investigated, such as cost or resource availability?

Author's response:

We thank the reviewer's high evaluation and valuable suggestions of our work. We have

expanded the conclusions upon the potential implications of this work. “This research not only provides a promising technique for the cost-effective and safe production of H₂ as well as the upgrading of biomass, but also establishes a foundation for exploring novel enol (such as phenol and catechol that have a low price and higher industrial value) electrooxidation coupled electrolysis systems or self-co-electrolysis^{52,53}.”

As suggested by the reviewer, we tried the electrochemical properties of other organic compounds with similar enol-like structures (ascorbic acid, phenol and catechol) (as shown in Fig. R6). We found that for the same electrolyte organic concentration with the same content of Fe in the single-atom catalyst, the performance of Fe catalyzed catechol was comparable to that of catalyzed AA at more than 1 A cm⁻², and 2.5% Fe@KJ exhibited ultra-high current densities in all three organic electrolytes, suggesting that this is common to enol-structured organics.

We know that AA is a natural biomass, which can be extracted from plants, is non-toxic and can be used as a carrier of hydrogen. Currently, the market price of AA is only \$3/kg, and the price of its product can reach \$400/kg. This indicates our hydrogen production systems from AA electrooxidation shows great potential for industrial application.

Fig. R6 LSV curves of Fe single-atom catalysts in different organic compounds.

Response to Reviewer #3:

This is an interesting and important work that reported the electrochemical oxidation of ascorbic acid (AA) to dehydroascorbic acid (DHA) and pairing it with hydrogen evolution reaction (HER) at the cathode for cogeneration of valuable chemical and H₂. The research includes electrochemical in-situ FTIR, and DFT, as well as flow cell tests. It can be published in Nature Communication after addressing the following comments.

Author's response:

We thank the reviewer's high evaluation and valuable suggestions. All the concerns raised from the reviewer have been addressed in detail as follows.

1. As we know, Fe³⁺ and AA can react to form DHA without a catalyst. Whether the 2.5%Fe@KJ reported by author can react with AA to produce DHA without electricity? Can the reaction between AA and Fe can selfpower the HER process like *Advanced Materials*, 2022, 34, 2200058; *Angewandte Chemie International Edition* 2023, e202218603.

Author's response:

Fig. R7 a, ¹H NMR characterization of oxidation products of AA without electricity. b, LSV of iron and platinum as the positive and negative electrodes in AA solution.

We thank the reviewers for their valuable suggestions. As suggested by the reviewer, we used 2.5% Fe@KJ as the anode and Pt as the cathode for the reaction without electricity for 5 h. By ¹H NMR characterization of the products, the NMR spectra only show the peaks of AA with negligible signal of DHA (Fig. R7a), indicating the 2.5%Fe@KJ can hardly react with AA to produce DHA without electricity. We assembled a self-co-electrolysis cell and tested its open-

circuit voltage was 0.08 V vs. Ag/AgCl and 0.013 V. vs. Ag/AgCl. We use iron plate and 2.5%Fe@KJ as the positive electrodes, and platinum as a negative electrode to test whether they can self-drive the HER process in the AA solution (Fig. R7b). From the polarization curves we can see that hydrogen precipitation occurs at a potential of -0.19 V and -0.39 V vs. Eoc at a current density of 10 mA cm⁻², indicating that the potential provided by itself cannot drive the occurrence of HER.

2. What is the stability of 2.5%Fe@KJ catalyst? The author should report the morphology and structure change of catalysts after 100 h durability tests which may also be important to understand the reaction mechanism.

Author's response:

We thank the reviewer for the valuable comment. The 2.5%Fe@KJ catalyst at 100 mA cm⁻² exhibits a remarkable stability. As proposed by reviewer's, we have done a morphology characterization with AC-HAADF-STEM after 100 hours. From the Fig. R8, we found that 2.5%Fe@KJ still exhibits atomic-level dispersion, which demonstrated the catalyst shows long-term stability.

Fig. R8 AC-HAADF-STEM images for catalysts after 100 h durability tests.

3. The yield of DHA is 87%. What are the other anode products?

Author's response:

We thank the reviewer for the valuable suggestion. By comparing the NMR peaks of standard AA and DHA, we found that the major product after 4 h was DHA (Fig. R9), and almost no peaks of other products were observed, indicating that the remaining species was unreacted AA.

To further verify our conclusion, we performed NMR characterization of the product reacted in the flow cell for 12 h at a potential of 0.65 V. Notably, the yield of DHA in this system can increase with time (Fig. R10). As the substrate concentration decreases, the conversion rate also decreases. We found that it achieved a DHA yield of over 98.3%. Evidently, the system can convert to a single DHA product without other by-products being generated (Fig. R10).

Fig. R9 a, NMR spectrum for AA with different oxidation time. b, Yield rates of DHA at different reaction times.

4. The quantitative analysis of anode products in the flow cell is suggested to be provided.

Author's response:

Thanks for reviewers' valuable comments. By ¹H NMR characterization, we remeasured the AA oxidation products after 12 h. With the increase of reaction time, it was found that the AA decreases gradually and the yield of DHA increases obviously. From Figs. R9 and R10, no new peaks were found, indicating that the anodic product after 12 h was DHA and the other organic species was AA.

Fig. R10 Fig.R10 ¹H NMR characterization and yield of anodic products with different operated times in flow cell under 0.65 V.

5. More electrochemical tests, such as ECSA, and so on, are suggested to be provided to solid the conclusion.

Author's response:

Thanks for reviewers' valuable comments. We have done C_{dl} and ECSA electrochemical tests (Figs. R11 and R12). First, we selected a non-Faraday voltage window (-0.05 to 0.05 V vs. Ag/AgCl) and performed CV cycling tests in this voltage range with sweep rates of (20, 40, 60, 80, 100, 120 mV/s). Next, points were taken at the potentials of 0 V vs. Ag/AgCl and the sweep speed was plotted as a function of the corresponding current density. Finally, the function is fitted linearly to find the slope of the function, which is also known as C_{dl} , and then the ECSA is calculated from C_{dl} .

The following equation was applied to measure the electrochemical surface area (ECSA) of the electrodes at different cell voltages.

$$C_T = I / (dE / dT) \quad (1)$$

$$ECSA = C_T / C^* A \quad (2)$$

where C_T is the total capacitance (F), I is the current (A), dE/dt is the voltage scan rate ($V s^{-1}$), C^* is the specific capacitance ($F cm^{-2}$), A refers to area of electrode and ECSA is the electrochemical surface area per geometric area ($cm^2 cm^{-2}$). The main limitation of this method is the general assumption that the specific capacitance of oxides is $60 \mu F cm^{-2}$, regardless of the oxide composition and crystalline structure, and without considering the composition of the electrolyte in which the measurement is carried out.

Fig. R11 C_{dl} for different metal single atoms and the corresponding ECSA.

Fig. R12 C_{dl} for different Fe content and the corresponding ECSA.

We found that the C_{dl} of different metal single-atom catalysts were around 15 mF cm^{-2} , and the corresponding ECSA were about 250. The above normalized electrocatalytic performance of ECSA showed that the 2.5% Fe@KJ single-atom catalyst exhibited better j_{ECSA} than the other catalysts. In addition, the normalized electrocatalytic performance of ECSA of different Fe contents also revealed that 2.5% Fe@KJ presented better catalytic activity than the other contents.

6. The size and format of the picture in the article are not uniform. The unit of current density is wrong (Figure 3a) and so on.

Author's response:

We sincerely appreciate the reviewer for notifying us of typographical errors. The size and format of the picture in the article have been modified. The unit of current density is rectified to be mA cm^{-2} in Fig. 3a.

7. Recent references on the electrooxidation of small organic molecules coupled with HER should be cited in this paper, especially under acidic conditions. Such as, *Angewandte Chemie International Edition* 2021, 60,21464–21472.

Author's response:

We thank the reviewers for their suggestions on our work. We have cited this reference in the corresponding places in the text. The cited references as follows:

“17 Li, Y., Wei, X., Han, S., Chen, L. & Shi, J. MnO_2 Electrocatalysts Coordinating Alcohol

Oxidation for Ultra-Durable Hydrogen and Chemical Productions in Acidic Solutions. *Angew. Chem. Int. Ed.* 60, 21464-21472 (2021).”

List of changes made to the manuscript

All changes in the manuscript are marked in blue

1. Updated parts of the introduction and conclusion, and have cited the references “*Angew. Chem. Int. Ed.* **n/a**, e202303563 (2023). *Adv. Energy Mater.* **13**, 2203455 (2023). *Angew. Chem. Int. Ed.* 60, 21464-21472 (2021). *Adv. Mater.* 34, 2200058 (2022). *Angew. Chem. Int. Ed.* 62, e202218603 (2023).”
2. Updated the image format and size, as well as the serial numbers and notes of the diagrams in the supporting information.
3. Added HRTEM with different Fe loadings and single-atom phase characterization of different metals in the characterization structures discussion part.
4. Added normalized ECSA and double layer capacitance to the description of performance section, and the figures are shown in SI.
5. Corrected the units of the vertical coordinate of Figure 3a.
6. Added the morphological structure of Fe catalyst after 100h of reaction in SI.
7. Corrected the peak of carbonyl group (1800 cm^{-1}) into 1797 cm^{-1} .
8. Added ^1H NMR characterization of the anodic product for 12 h in the flow cell in SI.
9. Added a note on the calculation of ECSA in the calculation method section.

REVIEWERS' COMMENTS

Reviewer #1 (Remarks to the Author):

In this revision, the authors gave more clarification by citing literature or using additional information on characterization and applications. The quality of this work is obviously improved. I suggest acceptance as it is.

Reviewer #2 (Remarks to the Author):

The authors have addressed the reviewers' comments. The revisions address the questions and provide additional insight regarding this innovative approach.

Reviewer #3 (Remarks to the Author):

I recommend its publication at the present form.